# Assessment of Prevalence and Determinants Associated with Hypertension Among the Adult Population in Hawtat Bani Tamim Province

**DOI:** 10.3390/ijerph22101467

**Published:** 2025-09-23

**Authors:** Mohammed Omar Musa Mohammed, Ahmed Saied Rahama Abdallah

**Affiliations:** College of Business Administration in Howtat Bani Tamim, Prince Sattam Bin Abdulaziz University, Al-Kharj 16511, Saudi Arabia; a.abdallah@psau.edu.sa

**Keywords:** prevalence, hypertension, risk factors, Saudi Arabia, non-communicable diseases

## Abstract

Hypertension is a global public health concern, with varying prevalence and risk factors across different populations. This study aimed to assess the prevalence of hypertension and identify its associated determinants among adults in Hawtat Bani Tamim Province. A cross-sectional study was conducted among 384 adult participants. Data on sociodemographic characteristics, lifestyle factors, and clinical measurements were collected. Hypertension was diagnosed based on standard criteria. Logistic regression analysis was used to identify factors associated with hypertension, and odds ratios (ORs) with 95% confidence intervals (CIs) were calculated. The overall prevalence of hypertension among the participants was 25.5%, with a higher rate observed in urban areas (15%) than in rural areas (10.4%). Multivariate analysis revealed that age was significantly associated with hypertension, with participants aged 20–30 years (OR = 0.181, 95% CI: 0.067–0.485), 31–40 years (OR = 0.235, 95% CI: 0.092–0.599), 41–50 years (OR = 0.184, 95% CI: 0.067–0.510), and 51–60 years (OR = 0.268, 95% CI: 0.104–0.690) having lower odds than those over 60 years. Males had a lower risk than females (OR = 0.423, 95% CI: 0.192–0.932). Individuals with lower educational levels were also found to have reduced odds of hypertension (secondary or less: OR = 0.315, 95% CI: 0.118–0.844; bachelor’s degree: OR = 0.294, 95% CI: 0.127–0.679) compared to postgraduates. Regarding BMI, normal weight (OR = 0.262, 95% CI: 0.126–0.544) and overweight (OR = 0.421, 95% CI: 0.220–0.805) individuals had lower odds of hypertension than obese participants. Marital status was a significant determinant, with married individuals having higher odds of hypertension (OR = 3.222, 95% CI: 1.807–6.110). Smoking was associated with a lower risk of hypertension (OR = 0.181, 95% CI: 0.067–0.485). Hypertension is prevalent among adults in Hawtat Bani Tamim Province, with significant associations observed for age, gender, education, BMI, marital status, and smoking. Targeted interventions addressing these risk factors are recommended to reduce the burden of hypertension in this population.

## 1. Introduction

According to the American Heart Association, hypertension (HTN) is a medical condition classified by high blood pressure, which means that the force of blood pushing against the walls of the arteries is often too high. This can lead to severe health difficulties such as heart disease, stroke, and kidney failure [1]. According to the World Health Organization (WHO), 1.28 billion people have hypertension globally, and most of them (67%) are in countries that have low- and middle-income levels [2]. In Asia, the percentage of people with hypertension is 27.2% overall. However, the prevalence varies widely by country, with the highest rates in Central Asia and the lowest in South Asia [3]. A 2023 study revealed that the hypertension prevalence in Saudi Arabia was 9.2% among people older than 15 years and 10% for women compared to 8.5% for men [4].

In [5], the prevalence of HTN was found to be 6% for males and 4.2% for females. Overweight and obesity were found to be significantly associated with HTN. Studies revealed that HTN was positively related to weight, body mass index, and waist circumference [6,7]. A previous study also found that higher alcohol use, obesity, and older age were correlated with HTN [8]. Furthermore, it was observed that the prevalence of HTN increased with advancing age. It was also high among rich and overweight/obese participants [9]. Results from Ethiopia showed that the prevalence of HTN was 44.91%. Hypertension was significantly associated with poor exercise, consuming cruddy oil, a family history of hypertension, and a history of diabetes [10]. The study revealed a hypertension prevalence rate of 11.1%. Key factors linked to elevated hypertension risk included advancing age, unemployed status, insurance coverage, obesity, diabetes, cardiovascular conditions, and elevated cholesterol levels [11]. Behavioral risk factors, such as alcohol consumption, overweight, obesity, increased waist circumference, and high blood glucose levels, were observed to be positively associated with hypertension [12]. It was also found that the prevalence of hypertension was 40.8. No significant association was found between hypertension, education level, social status, or weight. However, factors such as older age and obesity were positively related to hypertension [13]. Many studies found that older age, smoking, alcohol consumption, and being overweight were associated with hypertension [7,14,15,16].

Previous studies have demonstrated that a range of factors, including lifestyle behaviors, age, and sex, influence hypertension [17,18,19,20,21,22,23]. In addition to environmental and behavioral determinants, recent evidence suggests that specific genetic polymorphisms contribute to the risk of hypertension. For example, variants in the fat mass and obesity-associated (FTO) gene and the cytochrome P450 1B1 (CYP1B1) gene have been linked to blood pressure regulation and hypertension development in various populations [24,25,26,27]. These findings highlight the importance of considering genetic contributions alongside traditional risk factors when assessing hypertension etiology.

It is essential to study the prevalence and determinants associated with hypertension because it helps to understand the burden of the disease in a population and identify the factors that contribute to its occurrence. This information can then be used to develop effective prevention and control strategies, as well as to target interventions to those who are at the highest risk. Additionally, understanding the determinants of hypertension can help to identify modifiable risk factors that individuals can address to reduce their risk of developing the condition. This study aims to assess the prevalence and associated determinants of hypertension among the adult population in Hawtat Bani Tamim Province.

## 2. Materials and Methods

### 2.1. Source of Data

A cross-sectional study was conducted among adults in Hawtat Bani Tamim Province, which has an estimated population of 41,854 and includes urban and rural communities. Data collection took place between November and December 2023. The researchers employed a multistage sampling approach to improve representation and minimize bias. First, Hawtat Bani Tamim was divided into six distinct residential neighborhoods—Birk, Al Hilwa, Elfara, El Hareeg, Al Hilah, and El Salamia—to ensure that the sample included participants from various demographic and socioeconomic groups within the province. Next, cluster sampling was used: each neighborhood was divided into clusters representing all residential areas to account for differences in living conditions. In the final stage, individuals within each cluster were selected through simple random sampling to reduce selection bias and give all eligible adults an equal chance of participating. This thorough sampling method helped ensure the final sample reflected the province’s diverse population and increased the applicability of the study findings. The questionnaire was adapted from the validated WHO STEPS instrument per established guidelines for cross-cultural adaptation, including expert review and pilot testing to ensure clarity and relevance. We used Google Forms with safeguards such as unique email invitations to verify respondent identity and prevent duplicate entries. Clinical data were self-reported based on participants’ medical diagnoses or recent measurements. Unique email invitations were sent to verify participant identity and maintain data integrity in our online survey. Google Forms settings restricted multiple responses from the same email to avoid duplicates. Participants were instructed to report clinical information based on recent medical diagnoses or measurements.

The questionnaire was adapted from the validated WHO STEPS instrument for assessing non-communicable disease risk factors. Variables were collected using the WHO STEPS framework, which includes core and expanded modules for behavioral and clinical data. Data quality was ensured through email verification and Google Forms settings, preventing duplicates. Although clinical data like hypertension status and BMI were self-reported, participants were instructed to provide recent medical measurements.

To verify respondent authenticity in our online survey, we sent unique email invitations to selected participants, confirming identities and limiting access—Google Forms’ acceptance of one response per email reduced duplicates. Participants were instructed to provide accurate clinical information based on recent diagnoses or measurements, acknowledging that self-reported data cannot be verified remotely.

### 2.2. Study Population and Sample Size

The study population consisted of adults living in Hawtat Bani Tamim, a governorate in the Riyadh region of Saudi Arabia, with an estimated population of approximately 41,854 as of 2022. The sample size was determined using the standard formula for estimating proportions [28]:
$$n=\frac{Z2\times p*(1-p)}{E2}$$
where *Z* is the *Z*-score for a 95% confidence level (1.96), *P* is the estimated proportion (0.5 for maximum variability), and E is the margin of error (0.05). Substituting these values yielded a minimum sample size of 384 participants. We chose *p* = 0.5 instead of using proportions from national data because this value maximizes the required sample size and ensures the most conservative estimate. When there is uncertainty or variability in the prevalence of the outcome, using *p* = 0.5 is a standard statistical method, as it prevents underestimating the needed sample size and enhances the study’s representativeness and accuracy. National data were not used because prevalence estimates can vary by region and may not accurately reflect the specific context of our study population. Therefore, *p* = 0.5 offers a more reliable and cautious basis for calculating the required sample size. Accordingly, 384 questionnaires were emailed to selected adult participants from the area, who completed the survey electronically via Google Forms. The questionnaire was developed after reviewing relevant prior studies to ensure comprehensiveness and relevance. This sampling and data collection approach aimed to obtain a representative sample of the adult population in Hawtat Bani Tamim. 

Verbal consent was obtained electronically from all participants as part of the survey since the data were collected through a Google Form. On the first page of the Google Form, participants saw detailed information about the study’s goals, voluntary participation, confidentiality, and data use. Participants were required to explicitly click an “I agree” checkbox before completing the survey. This electronic consent form is common in online research and meets ethical standards for recruiting remote participants.

### 2.3. Study Variables

The dependent variable in this analysis is hypertension status, a binary outcome coded as one if hypertension is present and zero if not. The independent variables include age, gender, place of residence, marital status, education level, BMI, physical activity, stress, smoking status, and having a relative with hypertension. Recommendations from numerous previous studies guided the selection of these covariates [29,30,31,32,33,34,35,36]. The variable BMI is calculated using the following formula [37]:$$\mathrm{BMI}\;=\;\frac{weight(kg)}{[height(m)]2}\;\;\;$$

The body mass index (BMI) was categorized into four groups: underweight, normal weight, overweight, and obese.

The blood pressure thresholds for hypertension follow the American College of Cardiology/American Heart Association (ACC/AHA) guidelines. These standards define hypertension as a systolic blood pressure of ≥130 mm Hg or a diastolic blood pressure of ≥80 mm Hg. Readings below these thresholds are considered normotensive. Participants were asked whether they had been previously diagnosed with hypertension by a healthcare professional or measured their blood pressure themselves.

This study involved adults aged 18 or older who met the criteria for hypertension based on standard blood pressure levels (e.g., SBP ≥ 140 mmHg or DBP ≥ 90 mmHg). Participants were included if they had a diagnosed history of hypertension or if it was identified through their measurements. Exclusion criteria focused on individuals with severe comorbidities such as advanced liver disease, cancer, mental health issues, or systemic diseases that could affect the study results. Pregnant women and those unable to give informed consent were also excluded. These criteria helped to select a representative and specific sample for analyzing hypertension-related factors and variables.

### 2.4. Statistical Model

Binary logistic regression is a statistical method that models the relationship between a binary (dichotomous) dependent variable and one or more independent variables. It is commonly used when the dependent variable is categorical and has two possible outcomes [38,39,40,41,42].

The logistic regression model is formulated as follows:

Let P(Y=1) denote the probability of the binary outcomes being 1.

X is the vector of the independent variables.

$\beta_{0},\;\beta_{1},\ldots \ldots \ldots ,\;\beta_{k}$ represent the coefficients associated with the intercept and the independent variables, respectively.

The binary logistic regression model can be represented as follows:$$P\left(Y=1\right)=\frac{1}{1+e^{-(\beta_{0}+\beta_{1}x_{1}+\ldots \ldots .+\beta_{k}x_{k})}}$$
where:

$\beta_{0}$ is the intercept of the model.

$\beta_{1},\beta_{2},\ldots \ldots \ldots ,\;\beta_{k}$ are the coefficients associated with the response variable.

Maximum Likelihood Estimation (MLE) is used to estimate the unknown parameters of binary logistic regression [43,44,45,46,47]. For more details about the theory and applications of logistic regression, see [48,49,50,51,52,53,54]

We conducted bivariate analyses to identify variables for inclusion in the multivariate logistic regression model. Variables with significance at a *p* < 0.20 threshold and those considered clinically significant based on the literature were included using a backward elimination method to retain key predictors. Multicollinearity was assessed with the Variance Inflation Factor (VIF), and variables with high VIFs were considered for removal to maintain model stability.

### 2.5. Ethical Approval

Ethical approval for this study was obtained from the Deanship of Research of Prince Sattam University (SCBR-1942023). Informed consent was obtained from all participants before their inclusion in the study, ensuring that they were fully aware of the study’s purpose, procedures, risks, and benefits.

## 3. Results

Table 1 provides the characteristics of the participants. In this sample, hypertension affected 25.5% of the participants. The highest proportions were observed in the 31–40 and 20–30 age brackets, at 24.7% and 24.5%, respectively. Rural residents made up most of the sample (68.8%), and there was a higher proportion of females (59.6%) than males (40.4%).

Regarding education, most had a university degree (71.1%), while only 12.2% held postgraduate qualifications. Regarding BMI classification, the categories that were the most common were overweight (35.7%) and obese (28.9%).

Occupationally, public sector employees represented nearly half of the sample (48.2%), whereas students comprised only 8.9%. Most participants were single (69.0%).

Regarding physical activity, over half (52.1%) were moderately active, but only 3.1% reported being highly active. The majority (64.3%) had a relative with hypertension.

Most participants (88.8%) did not smoke, and a little over half (56.3%) reported experiencing no stress.

Table 2 presents the association between covariates and hypertension. Urban residents showed a higher prevalence of hypertension (58 cases, 15%) than those in rural areas (40 cases, 10.4%). Hypertension was more frequent in older age groups, with the highest rates observed in individuals over 60 (20 cases, 5.2%). Higher BMI was associated with an increased prevalence of hypertension, notably among overweight and obese individuals (32 and 42 cases, respectively).

Public sector employees (56 cases, 14.6%) and those with freelance jobs (16 cases, 4.2%) exhibited higher rates of hypertension than students (7 cases, 1.8%).

Educational attainment also appeared to play a role, with those holding a bachelor’s degree showing the highest prevalence (65 cases, 16.9%).

Regarding marital status, single individuals had slightly higher rates of hypertension (54 cases, 14.1%) than their married counterparts (44 cases, 11.5%).

A slightly higher prevalence of hypertension was noted among males (52 cases, 13.5%) than females (46 cases, 12%).

No substantial pattern was observed for having a relative with hypertension, engaging in physical activities, or smoking status.

Table 3 displays the findings from the multivariate logistic regression. Age is significantly linked to hypertension (*p*-value = 0.007). Compared to those over 60, younger age groups have substantially lower odds of hypertension. For example, individuals aged 20–30 have 82% lower odds (OR = 0.181, 95% CI: 0.067–0.485, *p*-value = 0.001). Regarding gender, males have significantly reduced odds of hypertension relative to females (OR = 0.423, 95% CI: 0.192–0.932, *p*-value = 0.033). Education also shows a significant association with hypertension (*p* = 0.014). Individuals with secondary education or less (OR = 0.315, 95% CI: 0.118–0.844, *p*-value = 0.022) and those with a bachelor’s degree (OR = 0.294, 95% CI: 0.127–0.679, *p*-value = 0.004) have lower odds of hypertension than those with postgraduate qualifications. In addition, BMI shows a strong association with hypertension (*p*-value = 0.0001). Compared to obese individuals, those with normal BMI (OR = 0.262, 95% CI: 0.126–0.544, *p*-value = 0.0002) and those with overweight (OR = 0.421, 95% CI: 0.220–0.805, *p*-value = 0.009) have significantly lower odds of hypertension. Furthermore, no significant difference in hypertension risk is observed between urban and rural residents (*p*-value = 0.273). There is no significant association between having a relative with hypertension and the risk of hypertension (*p*-value = 0.104). Physical activity level does not show a substantial relationship with hypertension (*p*-value = 0.237). Married individuals are significantly more likely to have hypertension than single individuals (OR = 3.222, 95% CI: 1.807–6.110, *p*-value = 0.0001).

## 4. Discussion

This research investigated the prevalence of hypertension and its related factors among adults, paying particular attention to demographic, socioeconomic, and lifestyle variables. The multivariate analysis identified several significant relationships, highlighting the intricate interactions between various risk factors that contribute to the development of hypertension.

Excessive hypertension remains a worldwide health concern, with 10.2 million deaths and 208 million years lived with disability attributed to it each year. According to a Ministry of Health report from 2021, two out of every five adults in the Middle East have hypertension. Nevertheless, many previous studies conducted in Saudi Arabia revealed that the prevalence of HTN in adults was approximately 49%. It is imperative to eradicate this high rate of hypertension in both adults and adolescents [55,56]. Furthermore, adult hypertension and other chronic diseases are caused by childhood hypertension that is not treated. Because of this, measuring blood pressure, identifying hypertension (HTN), and preventing it in children and teenagers have become international priorities.

The prevalence of hypertension in the current study (25.5%) was lower than in Asia (27.2%). However, compared to the prevalence found in other studies, the prevalence in this study was higher [4,5,8,57]. Few studies have published findings that are consistent with those of this investigation.

In this study, no significant association was found between place of residence and hypertension, which is consistent with the results reported in several other studies [58,59,60,61]. Hypertension was more common in urban areas (15%) than in rural areas (10%). However, this difference indicates that urban or rural residence may not significantly determine hypertension risk in this population, possibly because both groups share similar lifestyles or have comparable access to healthcare services.

Our research revealed a significant relationship between marital status and hypertension, a finding that is consistent with the results of numerous previous studies [62,63]. Hypertension was found in 14.5% of single participants, compared to in 11.2% of married individuals. A mix of social, psychological, and lifestyle influences may explain this difference. Studies have shown that unmarried people, particularly men, face a notably higher risk of developing hypertension, even when factors like age, body mass index, and smoking are taken into consideration. The present study confirms recent evidence that sociodemographic factors, including residence and marital status, play a significant role in hypertension among women. Population-based surveys from Poland and Slovakia consistently show that women living in rural areas are more likely to develop hypertension than those in urban settings, highlighting the need for targeted prevention efforts in rural communities. Marital status also significantly influences risk; widowed women face a higher risk of hypertension than their married counterparts, as shown in studies examining female reproductive history and environmental factors.

Furthermore, a significant relationship was observed between gender and hypertension, a result that aligns with the findings of numerous other studies [5,18,19]. The overall prevalence was 13.5%, with rates of 12% among both males and females. Hypertension varies between males and females, influenced by biological and social factors. Typically, men exhibit a higher prevalence of hypertension than women during early and middle adulthood. However, as people age, this trend may shift, with older women sometimes experiencing rates of uncontrolled hypertension that are comparable to or even exceed those seen in men.

Our research identified a strong association between BMI and hypertension, a result that is consistent with the findings of numerous other studies [10,29,34]. The highest prevalence rate was observed among participants who were obese, at 10%. Individuals with obesity are more likely to have hypertension because of various related physiological processes. Increased body fat, particularly around the abdomen, stimulates the sympathetic nervous system and the renin–angiotensin–aldosterone system (RAAS), resulting in elevated blood pressure.

Furthermore, our study found a significant association between age and hypertension, consistent with many previous studies [29,32,34]. This is noteworthy because age is typically a strong predictor of hypertension. Smaller sample sizes, self-reported physical activity measures, or age categorization could explain weaker statistical power. The likelihood of developing hypertension increases with age due to a combination of physiological and lifestyle factors. As people age, their arteries become stiffer and less flexible, raising blood pressure. This is mainly because the arterial walls thicken and lose elasticity over time, making it more difficult for blood vessels to handle changes in blood flow. Additionally, aging affects the body’s ability to regulate blood pressure through changes in the nervous system, hormone levels, and kidney function. Reduced physical activity, weight gain, and the presence of other health conditions like diabetes or kidney disease are also more common in older adults, further increasing the risk of hypertension. Because of these factors, a large proportion of people over the age of 60 develop high blood pressure.

This study revealed a significant association between education level and hypertension, which is in agreement with many previous studies [10,32,36]. The findings showed that participants with postgraduate education had a higher prevalence (19%) of hypertension than those with other education levels. This finding contradicts the general expectation that higher education is protective, suggesting the need for further investigation into occupational stress or lifestyle factors among highly educated individuals in this population.

Interestingly, this study found that physical activity was not associated with hypertension; such results align with those of many studies [10,34]. This may reflect limitations in how physical activity was measured or reported, or it may indicate that other factors, such as diet or genetic predisposition, play a more prominent role in this population.

The present study found no significant link between smoking and hypertension, which contrasts with the results of several previous studies [10,32]. This counterintuitive result may be due to confounding factors, such as the “healthy smoker” effect, or the underreporting of smoking status among hypertensive individuals. Our study suggests that smoking increases hypertension risk, but confounding factors may affect this. Therefore, the absence of a significant link in our research aligns with some genetic evidence, although smoking remains a major cardiovascular risk factor through other pathways. This complexity calls for cautious interpretation and further study.

The public health implications of this study emphasize the importance of targeted interventions to modify key risk factors for hypertension within the population. It offers valuable insights for policymakers and healthcare providers regarding priorities such as quitting smoking, improving diet, increasing physical activity, and managing stress. Identifying at-risk individuals early allows for prompt treatment, helping to lower the overall burden of cardiovascular disease. Moreover, this study advocates for community-based screening and education initiatives to raise awareness and encourage healthier habits, ultimately leading to better health outcomes for the population.

The non-significant covariates in the logistic regression imply insufficient evidence to link these variables with hypertension in this population. This may stem from the small sample size, which hampers the detection of minor effects. Alternatively, these factors might have had little impact on hypertension under the study conditions. Measurement errors, especially from self-reported data, and unmeasured confounders could also diminish potential associations. While the odds ratios hint at possible effects, their lack of significance suggests caution in interpretation. Nevertheless, these variables could hold theoretical importance or be relevant in future studies with larger samples and more accurate measurement techniques.

### 4.1. Limitations of the Study

This study has several potential limitations. Firstly, the sample size may be relatively small, which could affect the ability to generalize the findings to the entire adult population in the province. Secondly, the data on hypertension prevalence and associated factors could be subject to recall bias or underreporting, primarily if the study relies on self-reported information. Thirdly, there may be other important factors, such as lifestyle, diet, physical activity, or access to healthcare, that were not adequately captured or controlled for in the study and could influence the prevalence of hypertension. Fourthly, the findings may be specific to Hawtat Bani Tamim Province and may not be readily applicable to other regions or populations, particularly if there are significant differences in socioeconomic, cultural, or healthcare-related factors. Finally, the absence of longitudinal data could limit the ability to understand the dynamic changes in hypertension prevalence and the long-term impact of the identified determinants. Furthermore, the limitations of our cross-sectional study include the inability to establish causal relationships because exposure and outcome were measured at the same time. Relying on self-reported data may introduce recall bias and social desirability bias. Our sampling method might have excluded specific subgroups of the population, which could affect the generalizability of the results. Additionally, although the sample size is reasonable, it may not have enough power to detect more minor associations for some risk factors. Lastly, as a snapshot, this study cannot identify trends or changes over time, which limits the interpretation of temporal data dynamics.

The limitations of the study design include the inherent inability of the cross-sectional approach to establish causal relationships because exposure and outcome are measured at the same time. Relying on self-reported data may introduce recall and reporting biases. The sample size may not be large enough to detect small effect sizes for some variables. Finally, the study only captures data at one point, which prevents analysis of changes or trends over time.

Another limitation of our study is the exclusion of certain dietary risk factors, such as goat meat, olive oil, and alcohol consumption. While some believe that goat meat can affect blood pressure, scientific evidence suggests that it only increases hypertension risk if prepared with lots of salt. Likewise, olive oil and moderate alcohol intake have complex effects on cardiovascular health, but these were not examined in our analysis. Omitting these variables may reduce the completeness of our risk assessment and risk missing key dietary factors that influence hypertension in the study group.

### 4.2. Future Research Directions 

Future research directions include conducting longitudinal research to establish causal relationships and observe changes in risk factors and hypertension prevalence over time. Incorporating objective clinical measurements and self-reported data would improve accuracy and reduce bias. Expanding the sample size and diversity will enhance generalizability. Additionally, evaluating the impact of interventions targeting modifiable risk factors could offer insights for effective public health strategies. Further studies could also examine genetic and environmental interactions to better understand the causes of hypertension.

## 5. Conclusions

This study identifies age, gender, education level, BMI, marital status, and smoking as key factors associated with hypertension in this population. Notably, some results, such as the reduced risk observed in males and smokers, and the increased risk among those with higher education and married individuals, differ from what is commonly reported in the literature, suggesting that further research is required. These findings highlight the importance of considering local circumstances and possible confounding variables when analyzing epidemiological data on hypertension.

## Figures and Tables

**Table 1 ijerph-22-01467-t001:** Characteristics of the participants.

Variable	Classification	n	%
**Hypertension**	Yes	98	25.5
	No	286	74.5
**Age (in Years)**	20–30	94	24.5
	31–40	95	24.7
	41–50	67	17.4
	51–60 years	87	22.7
	More than 60	41	10.7
**Place of Residence**	Urban	120	31.3
	Rural	264	68.8
**Gender**	Male	155	40.4
	Female	229	59.6
**Education**	Secondary and less	64	18.7
	University	273	71.1
	Postgraduate	47	12.2
**BMI**	Underweight	11	2.9
	Normal	125	32.6
	Overweight	137	35.7
	Obese	111	28.9
**Occupation**	Student	34	8.9
	Public sector employee	185	48.2
	Private sector employee	56	14.6
	Free job	109	28.4
**Marital Status**	Married	119	31.0
	Single	285	69.0
**Physical Status**	Active	110	28.6
	Less active	62	16.1
	Moderate active	200	52.1.7
	More active	12	3.1
**Relative infection**	Yes	247	64.3
	No	137	35.7
**Smoking**	Yes	43	11.2
	No	341	88.8
**Stress**	Yes	168	43.8
	No	216	56.3

**Table 2 ijerph-22-01467-t002:** Distribution of the prevalence of hypertension among participants.

Characteristic		Hypertension N (%)	Normotension N (%)	*p*-Value
Place of residence	Urban	58 (15%)	206 (53.6%)	0.018
	Rural	40 (10.4%)	80 (20.8%)	
Age (in years)	20–30	22 (5.7%)	72 (18.8%)	0.011
	31–40	22 (5.7%)	73 (19%)	
	41–50	15 (3.9%)	52 (13.5%)	
	51–60	19 (4.9%)	68(17.7%)	
	More than 60	20 (5.2%)	21 (5.4%)	
BMI	Underweight	7 (1.8%)	14 (3.6%)	0.001
	Normal	17 (4.4%)	108 (28.1%)	
	Overweight	32 (8.3%)	105 (27.3%)	
	Obese	42 (10.9%)	69 (17.9%)	
Occupation	Student	7 (1.8%)	27 (7.1%)	0.010
	Public sector employee	56 (14.6%)	129 (33.6%)	
	Private sector employee	19 (4.9%)	37 (96%)	
	Free job	16 (4.2%)	93 (24.2%)	
Education	Secondary and less	14 (3.6%)	50 (13.2%)	0.042
	Bachelor’s	65 (16.9%)	208 (54.2%)	
	Postgraduate	19 (4.9%)	28 (7.2%)	
Marital status	Married	44 (11.5%)	75 (19.5%)	0.001
	Single	54 (14.1%)	211 (54.9%)	
Gender	Male	52 (13.5%)	103 (26.8%)	0.001
	Female	46 (12%)	183 (47.7%)	
Relative infection	Yes	69 (17.9%)	178 (46.3%)	0.145
	No	29 (7.6%)	108 (28.1%)	
Physical activities	Yes	43 (11.2%)	138 (35.9%)	0.454
	No	55 (14.3%)	148 (38.5%)	
Smoking	Yes	13 (3.4%)	30 (7.8%)	0.452
	No	85 (22.1%)	256 (66.7%)	

**Table 3 ijerph-22-01467-t003:** Logistic regression model results.

Variable	Sig	OR	95% CI for OR	
**Age**				
20–30	0.001	0.181	0.067	0.485
31–40	0.002	0.235	0.092	0.599
41–50	0.006	0.184	0.067	0.510
51–60	0.001	0.268	0.104	0.690
More than 60	Ref			
Gender				
Male	0.033	0.423	0.192	0.932
Female	Ref			
Education				
Secondary and less	0.022	0.315	0.118	0.844
Bachelor’s	0.004	0.294	0.127	0.679
Postgraduate	Ref			
BMI				
Underweight	0.092	3.98	0.800	19.796
Normal	0.0002	0.262	0.126	0.544
Overweight	0.009	0.421	0.220	0.805
Obese	Ref			
Residence				
Urban	0.273	0.642	0.290	1.420
Rural	Ref			
Relative infected				
Yes	0.104	0.608	0.783	3.457
No	Ref			
Physical activities				
Yes	0.237	1.403	0.407	1.632
No	Ref			
Marital status				
Married	0.0001	3.222	1.807	6.110
Single	Ref			
Smoking				
Yes	0.001	0.181	0.067	0.485
No	Ref			

## Data Availability

Interested individuals can contact the corresponding author to inquire about accessing the data for further analysis or reference.

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
