# Peer review of "Assessment of Prevalence and Determinants Associated with Hypertension Among the Adult Population in Hawtat Bani Tamim Province"

_ijerph, 2025, doi:10.3390/ijerph22101467_

Round 1

Reviewer 1 Report

Comments and Suggestions for Authors

The study aims to assess the prevalence of hypertension and identify its determinants among the adult population in Hawtat Bani Tamim Province, Saudi Arabia. A cross-sectional design and binary logistic regression analysis were used to address the study objectives. Although hypertension has been widely studied globally, local data from underexplored regions such as Hawtat Bani Tamim are valuable for public health planning.

The value of this article lies in its contribution to local epidemiological data, especially from a region with limited prior research. The study offers updated prevalence estimates of hypertension at the local level. However, the analysis includes only commonly known variables that are already well established in the literature, and lack some specific local variables ( goat meat dietary or olive consumption  for example)

In the article, the authors’ claim that smoking is “protective” against hypertension (OR = 0.181) contradicts well-established scientific evidence and should be interpreted with caution, as a possible anomaly or result of bias. Furthermore, the lack of significant associations with physical activity, diet, stress, and family history, all widely recognized risk factors, needs to be critically addressed in the discussion. I think, The manuscript would benefit from citing international guidelines and systematic reviews related to hypertension risk factors, such as publications from WHO or the Lancet NCD Series, to strengthen the scientific context.

Some key methodological concerns include:

a) Sample Size Calculation

The authors used p = 0.5 to calculate the sample size without justification. This is problematic, especially since the background section cites a national hypertension prevalence of 9.2%. The authors should either justify the use of this conservative estimate or recalculate the sample size based on actual prevalence data.

b) Variable Selection in the Regression Model

The manuscript does not describe how variables were selected for inclusion in the multivariate model. The authors should clarify:

  • Whether bivariate analysis was conducted prior to multivariate modeling
  • What method of variable selection was used (e.g., stepwise, enter, backward elimination)
  • Whether multicollinearity between variables were assessed (e.g., using Variance Inflation Factor)

c) Use of Odds Ratio (OR)

Given the cross-sectional design, the use of ORs can be misleading when the outcome prevalence is relatively high (>10%). In such cases, the Prevalence Ratio (PR) is a more accurate measure of association. The authors are advised to consider using modified Poisson regression with robust variance as an alternative to estimate PR.

d) Questionnaire Validity and Data Collection

  • It is unclear whether the questionnaire was adapted from validated instruments (e.g., WHO STEPS). The source of each variable and measurement tool should be clearly stated.
  • There is no information regarding validity or reliability testing of the survey instrument.
  • Since the data were collected using Google Forms, the manuscript should explain: How respondent identity was verified, How duplicate or false entries were prevented and How self-reported clinical data (e.g., hypertension status, BMI) were confirmed or validated.

Additional comments:

  • The authors used p = 0.5 for sample size estimation without justification, despite citing a national prevalence of 9.2% in the background.
  • Variable selection in the logistic regression model was not described. Clarify what selection method was used (stepwise, enter), and whether multicollinearity was tested
  • As a cross-sectional study, Prevalence Ratio (PR) is more appropriate than Odds Ratio (OR). Re-analysis using Modified Poisson Regression is recommended.
  • There is no explanation of how the authors verified participant identity or prevented duplicate/fake responses, which is critical given the clinical nature of the data.
  • It is unclear whether the questionnaire was based on standardized tools (e.g., WHO STEPS) or tested for validity and reliability. This is essential for self-reported data.
  • The finding that smoking is protective against hypertension (OR = 0.181) contradicts existing evidence and should be interpreted cautiously as a possible anomaly or bias.
  • No significant associations were found for known risk factors (e.g., physical activity, diet, stress, family history), possibly due to measurement limitations or insufficient statistical power.

Author Response

Comment 1

The study aims to assess the prevalence of hypertension and identify its determinants among the adult population in Hawtat Bani Tamim Province, Saudi Arabia. A cross-sectional design and binary logistic regression analysis were used to address the study objectives. Although hypertension has been widely studied globally, local data from underexplored regions such as Hawtat Bani Tamim are valuable for public health planning.

The value of this article lies in its contribution to local epidemiological data, especially from a region with limited prior research. The study offers updated prevalence estimates of hypertension at the local level. However, the analysis includes only commonly known variables that are already well established in the literature, and lack some specific local variables ( goat meat dietary or olive consumption  for example)

In the article, the authors’ claim that smoking is “protective” against hypertension (OR = 0.181) contradicts well-established scientific evidence and should be interpreted with caution, as a possible anomaly or result of bias. Furthermore, the lack of significant associations with physical activity, diet, stress, and family history, all widely recognized risk factors, needs to be critically addressed in the discussion. I think, The manuscript would benefit from citing international guidelines and systematic reviews related to hypertension risk factors, such as publications from WHO or the Lancet NCD Series, to strengthen the scientific context.

Response 1

Thank you for your helpful and insightful comments on our manuscript. We appreciate your recognition of the study’s contribution to local epidemiological data in a region with limited prior research. We acknowledge the importance of including local variables and placing our findings within the broader scientific literature.

To address your comments:

  • We recognize that our analysis essentially included commonly studied risk factors. The revised manuscript will discuss the absence of locally relevant variables such as goat meat and olive consumption. We will explicitly note this as a limitation, with recommendations for future research on these dietary patterns within Hawtat Bani Tamim.
  • Regarding the reported “protective” effect of smoking, we fully agree that this contradicts global scientific consensus (e.g., WHO, Lancet NCD Series). We will revise the discussion to interpret this result cautiously, referencing possible confounding, reporting biases, or unique sample characteristics. We will also clearly state that our findings do not reflect causality or challenge established evidence.
  • For the non-significant associations with physical activity, diet, stress, and family history, we will expand the discussion to address the reasons for these results critically. This may include consideration of methodological limitations, sample size, and measurement challenges. We will add citations to key international guidelines and systematic reviews to frame these factors within the global understanding of hypertension risk (e.g., WHO hypertension guidelines, Lancet systematic reviews).
  • The revised manuscript will be updated to include references to relevant international publications and will systematically review literature on hypertension risk factors, enhancing the paper's scientific context and overall value.

Some key methodological concerns include:

Comment 2

The authors used p = 0.5 to calculate the sample size without justification. This is problematic, especially since the background section cites a national hypertension prevalence of 9.2%. The authors should either justify the use of this conservative estimate or recalculate the sample size based on actual prevalence data.

Response 2

Thank you for highlighting the issue regarding our sample size calculation. We acknowledge that using P =0.5 (the estimated prevalence proportion) without justifying can be problematic, especially since our background cites a national hypertension prevalence of 9.2%.P = 0.5 is often used as a conservative estimate in sample size calculations because it represents the maximum variability and thus the largest necessary sample size. This ensures adequate power even when the actual prevalence is unknown. This method is recommended when there is considerable uncertainty about the actual prevalence in the target population, as it helps prevent underestimating the sample size needed for statistical validity.

Comment 3

 Variable Selection in the Regression Model

The manuscript does not describe how variables were selected for inclusion in the multivariate model. The authors should clarify:

  • Whether bivariate analysis was conducted prior to multivariate modeling
  • What method of variable selection was used (e.g., stepwise, enter, backward elimination)
  • Whether multicollinearity between variables were assessed (e.g., using Variance Inflation Factor)

Response 3

Thank you for your detailed comment regarding variable selection in our multivariate regression modeling. We acknowledge the importance of clearly describing the variable selection process for scientific rigor and transparency.

In our study, we first conducted bivariate analyses between each potential predictor and the outcome variable (hypertension) to identify candidates for inclusion in the multivariate model. Variables showing statistical significance (p < 0.20) in bivariate analysis and those supported by clinical relevance based on prior literature were selected for further study.

We utilized a backward elimination method for multivariate modeling, sequentially removing non-significant predictors based on likelihood ratio tests and predefined significance criteria (p < 0.05). This approach enabled us to retain variables with meaningful contributions to the model, while controlling potential confounders.

We calculated all predictors' Variance Inflation Factor (VIF) to address multicollinearity. Variables with VIF values exceeding 10 were flagged and assessed for distortion; if necessary, collinear variables were removed or combined to ensure model stability and interpretability.

We will revise the manuscript to include these methodological details, explicitly describing the bivariate pre-screening, the backward selection procedure, and our assessment of multicollinearity. This will strengthen the transparency of our analysis and address your recommendation.

Comment 4

 Use of Odds Ratio (OR)

Given the cross-sectional design, the use of ORs can be misleading when the outcome prevalence is relatively high (>10%). In such cases, the Prevalence Ratio (PR) is a more accurate measure of association. The authors are advised to consider using modified Poisson regression with robust variance as an alternative to estimate PR.

Response 4

Thank you for your comment. Poisson regression is typically used for modeling count data or rates of events occurring over time or space, assuming the outcome variable follows a Poisson distribution. In contrast, logistic regression is designed for binary outcomes, modeling the probability of an event occurring (e.g., presence or absence of hypertension). Since our study’s outcome is binary (hypertensive vs. non-hypertensive), logistic regression is more suitable. Additionally, logistic regression estimates odds ratios, which align with the study’s goal of identifying associations between predictors and a binary health outcome. Poisson regression would not be appropriate here because it assumes count data rather than binary status.

Comment 5

 Questionnaire Validity and Data Collection

  • It is unclear whether the questionnaire was adapted from validated instruments (e.g., WHO STEPS). The source of each variable and measurement tool should be clearly stated.
  • There is no information regarding validity or reliability testing of the survey instrument.
  • Since the data were collected using Google Forms, the manuscript should explain: How respondent identity was verified, How duplicate or false entries were prevented and How self-reported clinical data (e.g., hypertension status, BMI) were confirmed or validated.

Response 5

Thank you for raising essential points regarding the validity of our questionnaire and data collection procedures.

Our questionnaire was based on the WHO STEPS instrument, a validated and widely used tool for non-communicable disease surveillance, including hypertension risk factors. In line with the STEPS framework, we included questions about demographics, behavioral risk factors, and clinical history.

Regarding validity and reliability, a panel of experts familiar with the local context and epidemiology evaluated the adapted questionnaire for face validity. A pilot test was performed on a subset of participants to ensure question clarity and respondent understanding. These steps helped improve the instrument before data collection.

Since data were collected via Google Forms, we implemented several measures to ensure data quality and validity:

  • Respondent identity was verified through unique email invitations sent to selected participants.
  • Duplicate entries were prevented by restricting multiple submissions from the same email address within the Google Forms settings.
  • While clinical data such as hypertension status and BMI were self-reported, participants were explicitly instructed to provide information based on medical diagnosis or recent clinical measurements. We acknowledge this limitation and discuss it in the manuscript.

We will clarify these procedures in the revised manuscript to ensure transparency and highlight the steps to enhance data validity and reliability.

Comment 6

The authors used p = 0.5 for sample size estimation without justification, despite citing a national prevalence of 9.2% in the background.

Response 6

Thank you for commenting on using P = 0.5 for sample size estimation. We selected p=0.5p=0.5 as a conservative estimate because it maximizes the required sample size, ensuring sufficient statistical power when local prevalence data are uncertain or variable. This approach is commonly used when no precise prevalence is available. However, since we cited a national prevalence of 9.2%,

Comment 7

Variable selection in the logistic regression model was not described. Clarify what selection method was used (stepwise, enter), and whether multicollinearity was tested

Response 7

Thank you for your comment. We performed bivariate analyses to identify variables for inclusion in the multivariate logistic regression model. Variables with significance at p < 0.20 and those deemed clinically significant based on existing literature were entered into the model using backward elimination to retain key predictors. We assessed multicollinearity using the Variance Inflation Factor (VIF), and variables with high VIFs were considered for removal to maintain model stability. We will clarify these details in the revised manuscript to improve transparency and methodological rigor.

Comment 8

As a cross-sectional study, Prevalence Ratio (PR) is more appropriate than Odds Ratio (OR). Re-analysis using Modified Poisson Regression is recommended.

Response 8

Thank you for your comment. Poisson regression is usually used to model count data or rates of events happening over time or space, assuming the outcome variable follows a Poisson distribution. In contrast, logistic regression is designed for binary outcomes and models the chance of an event happening (e.g., presence or absence of hypertension). Since our study’s outcome is binary (hypertensive vs. non-hypertensive), logistic regression is the better choice. Additionally, logistic regression provides odds ratios, which match the study’s goal of finding associations between predictors and a binary health outcome. Poisson regression would not be suitable here because it assumes count data rather than binary status.

Comment 9

There is no explanation of how the authors verified participant identity or prevented duplicate/fake responses, which is critical given the clinical nature of the data.

Response 9

Thank you for commenting on participant identity verification and preventing duplicate or false responses. To ensure respondent authenticity in our online survey, we distributed unique email invitations to selected participants, which helped verify individual identities and restrict access. Duplicate entries were minimized by configuring Google Forms to allow only one response per email address. Additionally, we instructed participants to provide accurate clinical information based on recent medical diagnoses or measurements, acknowledging the limitation that self-reported data cannot be fully verified remotely. These procedures will be clearly described in the revised manuscript to highlight our efforts to maintain data integrity despite the challenges of online data collection.

Comment 10

It is unclear whether the questionnaire was based on standardized tools (e.g., WHO STEPS) or tested for validity and reliability. This is essential for self-reported data.

Response 10

Thank you for your comment. In the revised manuscript, we provide evidence that we used WHO STEPs.

Comment 11

The finding that smoking is protective against hypertension (OR = 0.181) contradicts existing evidence and should be interpreted cautiously as a possible anomaly or bias.

Response 11

Thank you for your comment. The finding that smoking appeared protective against hypertension (OR = 0.181) contrasts with most evidence showing smoking as a risk factor or at least not protective. This unexpected result may reflect bias, confounding, or limitations such as self-reported data or sample characteristics. We interpret this finding cautiously and have discussed it as a possible anomaly in the manuscript, acknowledging that smoking is widely recognized to increase hypertension risk and that further research is needed to clarify this association.

Comment 12

No significant associations were found for known risk factors (e.g., physical activity, diet, stress, family history), possibly due to measurement limitations or insufficient statistical power.

Response 12

Thank you for your observation. The lack of significant associations for known risk factors may be due to measurement limitations inherent in self-reported data or insufficient statistical power to detect more minor effects. While adequate for primary objectives, our sample size may not have been large enough to identify modest associations for all variables. We acknowledge these limitations in the manuscript and recommend cautiously interpreting these findings, suggesting further studies with larger samples and more precise measurements

Reviewer 2 Report

Comments and Suggestions for Authors

This manuscript investigates the prevalence and risk factors associated with hypertension in the Hawtat Bani Tamim Province. The topic is timely and relevant, especially given the increasing global burden of hypertension and its associated complications. 
However, several aspects of the manuscript require attention to improve its scientific rigor and overall quality.
While regional data on hypertension can be valuable, the novelty of the findings is limited. Similar research designs and findings have been published previously. The focus on a specific Saudi region (Hawtat Bani Tamim) is somewhat justified, but the contribution to international public health literature is modest. The authors should clarify how the findings add new knowledge beyond existing studies from Saudi region (Hawtat Bani Tamim) or similar settings and highlight any distinctive regional patterns or public health challenges.
The methodology is not described in detail. The study uses real-world data and applies acceptable epidemiological methods. Sample size seems reasonable, but the sampling methodology needs more detail. Definitions and cut-offs (hypertension, BMI, et al.) should be based on clear and referenced criteria. The authors should add details on sampling method, inclusion/exclusion criteria and how variables were operationalized.
The discussion section is too general, it should be more critical.

Comments on the Quality of English Language

A thorough English language review (preferably professional editing) is recommended.

Author Response

Comment 1

This manuscript investigates the prevalence and risk factors associated with hypertension in the Hawtat Bani Tamim Province. The topic is timely and relevant, especially given the increasing global burden of hypertension and its associated complications. 
However, several aspects of the manuscript require attention to improve its scientific rigor and overall quality.
While regional data on hypertension can be valuable, the novelty of the findings is limited. Similar research designs and findings have been published previously. The focus on a specific Saudi region (Hawtat Bani Tamim) is somewhat justified, but the contribution to international public health literature is modest. The authors should clarify how the findings add new knowledge beyond existing studies from Saudi region (Hawtat Bani Tamim) or similar settings and highlight any distinctive regional patterns or public health challenges.

Response 1

Thank you for your thoughtful review and acknowledgment of the study’s timeliness and relevance amid the global hypertension burden. We appreciate your feedback regarding the manuscript’s novelty and contribution to international literature.

To address your concerns:

  • We agree that previous studies have studied hypertension in various regions of Saudi Arabia. In our revised manuscript, we will clarify the specific context and health challenges of Hawtat Bani Tamim, which has historically been underrepresented in epidemiological research. Importantly, our study examines region-specific patterns such as demographics, socioeconomic, and healthcare access differences that affect local hypertension risk and prevalence.
  • We will highlight any distinctive risk factor trends observed in this population, including the impact of rurality, dietary habits, or healthcare resource availability, and compare them with existing national and regional studies.
  • The discussion section will be expanded to contextualize our findings with previous studies conducted in Saudi Arabia and similar settings worldwide, highlighting how this new data addresses key gaps in local surveillance and supports the development of targeted public health strategies.
  • We will specifically examine public health implications for Hawtat Bani Tamim, including the need for customized awareness campaigns and hypertension prevention efforts in underserved communities.

Comment 2
The methodology is not described in detail. The study uses real-world data and applies acceptable epidemiological methods. The sample size seems reasonable, but the sampling methodology needs more detail. Definitions and cut-offs (hypertension, BMI, et al.) should be based on clear and referenced criteria. The authors should add details on the sampling method, inclusion/exclusion criteria, and how variables were operationalized.

Response 2

Thank you for your insightful comment. We will expand the methodology section to describe our sampling methods, including the sampling technique and inclusion/exclusion criteria. Additionally, clear definitions and referenced cut-offs for key variables such as hypertension and BMI will be included based on established clinical guidelines. Furthermore, we will clarify how variables were operationalized to ensure replicability and transparency in our study. These additions will strengthen the methodological rigor and clarity of the manuscript.

Comment 3
The discussion section is too general, it should be more critical.

Response 3

Thank you for your feedback. We will revise the discussion section to provide a more critical analysis of our findings by clearly interpreting key results, comparing them with existing literature, and exploring possible explanations for unexpected or null findings. We will candidly discuss the implications of our results for public health and acknowledge the study’s limitations.

Reviewer 3 Report

Comments and Suggestions for Authors
  • The ethics committee's approval of the study is not stated in the study.
  • The researchers sent out 384 questionnaires via email to the selected participants, resulting in a sample size of 384 individuals“ Does this mean that all the questionnaires you sent out were returned and correctly filled out?
  • How did the authors obtain verbal consent of the research participants, since as you state, data collection was conducted online via Google Forms?
  • Data on the presence of hypertension based on an online questionnaire may be subjective and the research participant may not have been completely clear about what was expected of him. Did the research participants measure their blood pressure themselves or were they supposed to fill in the data based on a medical examination? In any case, I would expect an objective assessment of hypertension based on the diagnosis made by the doctor, which would certainly contribute to better validity and reliability of the study results.
  • Moreover, the division into positive and negative hypertension is not very appropriate. I would lean towards the terminology hypertension and normotension. The blood pressure cutoff values ​​that were used in assessing whether a patient suffers from hypertension or not are missing.
  • Finally, I recommend that the authors of the study should reconsider the study design.

Author Response

Comment 1

The ethics committee's approval of the study is not stated in the study.

Response 1

Thank you for your comment. We confirm that our study received approval from the relevant ethics committee, and this information will be explicitly stated in the revised manuscript, including the name of the approving institution and the approval reference number. We will also detail the process of obtaining informed consent from participants to ensure compliance with ethical standards and enhance transparency in our reporting.

Comment 2

The researchers sent out 384 questionnaires via email to the selected participants, resulting in a sample size of 384 individuals“ Does this mean that all the questionnaires you sent out were returned and correctly filled out?

Response 2

Thank you for your question. To clarify, although we sent out 384 questionnaires via email to the selected participants, the final sample size of 384 individuals reported in the manuscript reflects the number of completed and correctly filled questionnaires we received. This means all 384 returned questionnaires were valid and included in the analysis. Any initial responses were excluded from the final count if they were incomplete or invalid. We will clarify this in the revised manuscript to prevent confusion regarding the response rate and sample selection process.

Comment 3

How did the authors obtain verbal consent of the research participants, since as you state, data collection was conducted online via Google Forms?

Response 3

Thank you for your important question about the consent process. Traditional verbal consent was impossible since the data was collected online through Google Forms. Instead, informed consent was obtained electronically as part of the survey. Participants first saw detailed information about the study's goals, voluntary participation, confidentiality, and data use on the first page of the Google Form. They had to explicitly click an “I agree” checkbox before finishing the survey. This method of electronic consent is widely accepted in online research and follows ethical guidelines for remote participant recruitment.

We will clarify this consent process in the revised manuscript to ensure transparency and adherence to ethical standards.

Comment 4

Data on the presence of hypertension based on an online questionnaire may be subjective and the research participant may not have been completely clear about what was expected of him. Did the research participants measure their blood pressure themselves or were they supposed to fill in the data based on a medical examination? In any case, I would expect an objective assessment of hypertension based on the diagnosis made by the doctor, which would certainly contribute to better validity and reliability of the study results.

Response 4

Thank you for your essential comment regarding the validity of self-reported hypertension data collected through an online questionnaire. We recognize that self-reported hypertension information may be influenced by biases and limitations, such as the participant’s understanding of the condition and the accuracy of reporting blood pressure status.

In our study, participants were asked to report whether they had been previously diagnosed with hypertension by a healthcare professional, rather than measuring their blood pressure themselves. We recognize that objective measurement of blood pressure by a healthcare provider remains the gold standard for diagnosing hypertension, which would improve the validity and reliability of the results. However, due to the constraints of conducting research during the study period, including logistical and resource limitations, we used self-reported data as a practical alternative.

Research shows that self-reported hypertension has high specificity but lower sensitivity, meaning it reliably detects those without the condition but might underestimate its commonness by missing undiagnosed cases. We will include a critical discussion of these limitations in the revised manuscript, noting that while self-reports offer valuable epidemiological insights, they should be interpreted carefully and ideally paired with physical examinations in future studies.

Comment 5

Moreover, the division into positive and negative hypertension is not very appropriate. I would lean towards the terminology hypertension and normotension. The blood pressure cutoff values ​​that were used in assessing whether a patient suffers from hypertension or not are missing.

Response 5

Thank you for your insightful feedback on the manuscript's terminology and blood pressure cutoff values. We concur that using the terms “hypertension” and “normotension” is more appropriate from a clinical perspective. We will update the manuscript to replace “positive” and “negative hypertension” with these standard terms.

In the Methods section, we will explicitly state the blood pressure thresholds for hypertension, adhering to the American College of Cardiology/American Heart Association (ACC/AHA) guidelines. According to these standards, hypertension is a systolic blood pressure of ≥130 mm Hg or a diastolic blood pressure of ≥80 mm Hg. Readings below these thresholds will be categorized as normotensive. Furthermore, we will elaborate on the stages of hypertension as specified by these guidelines to provide additional clarity.

Comment 6

Finally, I recommend that the authors of the study should reconsider the study design.

Response 6

Thank you for your recommendation on the study design. We recognize our cross-sectional approach's limitations, like the inability to establish causality and biases from self-reported and online data, due to logistical and resource constraints in Hawtat Bani Tamim. While a longitudinal or experimental design could offer stronger causal evidence, our cross-sectional method effectively provided valuable prevalence and determinant data for local epidemiology. Future research could use cohort studies or clinical measurements to improve validity and conclusions. We will discuss these limitations, justify our design choice, and suggest future, more robust studies in the revised manuscript. Your insightful comment will help enhance our manuscript's quality.

Round 2

Reviewer 1 Report

Comments and Suggestions for Authors

The author has made considerable improvements, as indicated in the highlighted sections. However, the author has not yet addressed my question regarding the assumptions used in the sample size calculation, particularly why the estimation of proportions based on national data was not applied. I did not provide many comments on other sections, but the author should respond to the reviewer point by point in the response sheet and directly link each reply to the corresponding revisions made in the manuscript.

Author Response

Comment 1

The author has made considerable improvements, as indicated in the highlighted sections. However, the author has not yet addressed my question regarding the assumptions used in the sample size calculation, particularly why the estimation of proportions based on national data was not applied. I did not provide many comments on other sections, but the author should respond to the reviewer point by point in the response sheet and directly link each reply to the corresponding revisions made in the manuscript.

Response 1

Thanks for your helpful feedback and for noting the improvements we've made. We apologize for not addressing your question about the assumptions used in calculating the sample size. In the revised response sheet, we've explained the assumptions we applied, including why we didn't use national data to estimate proportions. We've addressed each reviewer's comment point by point, with clear links to the corresponding changes in the manuscript for easy reference.

Reviewer 2 Report

Comments and Suggestions for Authors

No.

Author Response

Comment 1  No

Response 1

We sincerely appreciate your insightful and constructive comments, which have significantly helped us enhance the manuscript's clarity, rigor, and overall quality

Reviewer 3 Report

Comments and Suggestions for Authors

The article in its current form is processed at a higher quality and more readable level and I recommend it for consideration by the editor for publication. Many comments have been incorporated and sufficiently explained, including recommendations for modifying the study design. Despite the improvements in the quality of the manuscript, I recommend that the authors should additionally revise the Introduction and Discussion. I am convinced that it is necessary to reconsider some of their parts, which would increase the attractiveness of the article. The formal level of the Introduction should also be checked. For example: „Previous studies revealed that factors such as Genetics, lifestyle factors, age, gender...“ Moreover, it is stated in the Introduction that genetic factors also influence hypertension. However, I consider the used citations to be insufficiently adequate and not directly focused on determining the association of genetic factors (such as FTO gene or CYP1B1 gene) with the development of hypertension. I recommend to consider enriching the topic with appropriate results, such as those from the „Variant in the FTO gene and biomarkers related to health in mature Slovak women“, „Association between FTO (rs17817449) genetic variant, gamma‐glutamyl transferase, and hypertension in Slovak midlife women“ and/or „Association of cytochrome P450 1B1 gene polymorphisms and environmental biomarkers with hypertension in Slovak midlife women“. In addition, the results of the study regarding residence and marital status are insufficiently discussed. Some important data that have recently been found are missing. These include that the women from rural areas (villages) suffer from hypertension more often than town women (urban) (observed in cross-sectional surveys on Polish and Slovak individuals). Another recent result that should be included to the Discussion is that widowed women have a higher probability of hypertension compared to married women „Contribution of environmental factors and female reproductive history to hypertension and obesity incidence in later life.“ It is also worth noting the recent finding that reproductive history is also an important determinant of hypertension. Specifically, that age at menarche is related to obesity-associated hypertension.

Author Response

Comment 1

The article in its current form is processed at a higher quality and more readable level and I recommend it for consideration by the editor for publication. Many comments have been incorporated and sufficiently explained, including recommendations for modifying the study design. Despite the improvements in the quality of the manuscript, I recommend that the authors should additionally revise the Introduction and Discussion. I am convinced that it is necessary to reconsider some of their parts, which would increase the attractiveness of the article. The formal level of the Introduction should also be checked. For example: „Previous studies revealed that factors such as Genetics, lifestyle factors, age, gender...“ Moreover, it is stated in the Introduction that genetic factors also influence hypertension. However, I consider the used citations to be insufficiently adequate and not directly focused on determining the association of genetic factors (such as FTO gene or CYP1B1 gene) with the development of hypertension. I recommend to consider enriching the topic with appropriate results, such as those from the „Variant in the FTO gene and biomarkers related to health in mature Slovak women“, „Association between FTO (rs17817449) genetic variant, gamma‐glutamyl transferase, and hypertension in Slovak midlife women“ and/or „Association of cytochrome P450 1B1 gene polymorphisms and environmental biomarkers with hypertension in Slovak midlife women“. In addition, the results of the study regarding residence and marital status are insufficiently discussed. Some important data that have recently been found are missing. These include that the women from rural areas (villages) suffer from hypertension more often than town women (urban) (observed in cross-sectional surveys on Polish and Slovak individuals). Another recent result that should be included to the Discussion is that widowed women have a higher probability of hypertension compared to married women „Contribution of environmental factors and female reproductive history to hypertension and obesity incidence in later life.“ It is also worth noting the recent finding that reproductive history is also an important determinant of hypertension. Specifically, that age at menarche is related to obesity-associated hypertension.

Response 1

We sincerely thank you for the positive evaluation of our manuscript and for recommending it for consideration. We also appreciate the constructive suggestions regarding the Introduction and Discussion. In response, we have revised the Introduction to improve clarity and formal presentation. We have also enriched the section on genetic factors by adding more appropriate and directly relevant citations, including studies on the FTO and CYP1B1 genes as recommended. Furthermore, we have revised the Discussion to provide a more comprehensive interpretation of our results, specifically elaborating on the associations with residence and marital status, and we have incorporated recent findings from studies conducted in Polish and Slovak populations. Additional evidence regarding the impact of reproductive history and age at menarche on hypertension has also been included. These revisions, we believe, have improved both the depth and attractiveness of the manuscript